# m^1^A RNA Modification in Gene Expression Regulation

**DOI:** 10.3390/genes13050910

**Published:** 2022-05-19

**Authors:** Hao Jin, Chunxiao Huo, Tianhua Zhou, Shanshan Xie

**Affiliations:** 1The Children’s Hospital, Zhejiang University School of Medicine, National Clinical Research Center for Child Health, Hangzhou 310052, China; hawkim@zju.edu.cn; 2Department of Cell Biology, Zhejiang University School of Medicine, Hangzhou 310058, China; 12018074@zju.edu.cn; 3Cancer Center, Zhejiang University, Hangzhou 310058, China

**Keywords:** *N*^1^-methyladenosine(m^1^A), RNA modification, gene expression

## Abstract

*N*^1^-methyladenosine (m^1^A) is a prevalent and reversible post-transcriptional RNA modification that decorates tRNA, rRNA and mRNA. Recent studies based on technical advances in analytical chemistry and high-throughput sequencing methods have revealed the crucial roles of m^1^A RNA modification in gene regulation and biological processes. In this review, we focus on progress in the study of m^1^A methyltransferases, m^1^A demethylases and m^1^A-dependent RNA-binding proteins and highlight the biological mechanisms and functions of m^1^A RNA modification, as well as its association with human disease. We also summarize the current understanding of detection approaches for m^1^A RNA modification.

## 1. Introduction

Cellular RNAs contain more than 170 different types of chemical modifications across species [1]. *N*^1^-methyladenosine(m^1^A) is a reversible methylation involving the addition of a methyl group at the *N*^1^ position of adenosine in cellular transcripts [2]. The methyl group can block the normal Watson–Crick base pairing of A:T or A:U, resulting in an unstable mismatch with other nucleosides by forming Hoogsteen base pairs [3]. The secondary structure and RNA–protein interaction of m^1^A-modified RNAs are also altered under physiological conditions [4]. As a dynamic and reversible post-transcriptional RNA modification, m^1^A can be installed by methyltransferases, removed by demethylases and recognized by m^1^A-dependent RNA-binding proteins [2,5]. m^1^A RNA modification affects RNA metabolism, including RNA structure, stability and mRNA translation, thereby regulating gene expression and several fundamental cellular processes [6].

m^1^A RNA modification has been found with high abundance in transfer RNAs (tRNAs) and ribosomal RNAs (rRNAs) but at low levels in messenger RNAs (mRNAs) [7,8,9,10,11,12]. It occurs in the tRNA of bacteria, archaea and eukaryotes at positions 9, 14, 16, 22, 57 and 58 (m^1^A9, m^1^A14, m^1^A16, m^1^A22, m^1^A57, and m^1^A58, respectively) [13]. In cytosolic (cyt) tRNAs, m^1^A RNA modification occurs at five different positions (9, 14, 22, 57, and 58) [14,15]. Among them, m^1^A14 has only been identified in cyt(tRNA)^Phe^ from mammals, m^1^A22 has only been identified in bacteria tRNAs, and m^1^A57 has been identified in archaea existing only transiently as an intermediate of 1-methylinosine (m^1^I) [14,15]. In mitochondria, m^1^A9 is quite abundant and found in 14 species of mt-tRNA, while m^1^A58 is a minor modification with a 17% frequency found in four species of mt-tRNAs [16]. Additionally, m^1^A16 is unique to human mt-tRNA^Arg^, and its frequency is approximately 20% [16]. For rRNAs, the nuclear-encoded large subunit rRNA m^1^A645 in 25S rRNA and m^1^A1322 in 28S rRNA located in the peptidyl transfer center of the ribosome are conserved in budding yeast and humans, respectively [17,18,19], and m^1^A is conserved at position 947 of 16S rRNA in the mitochondrial ribosome of vertebrates [20]. Regarding mRNAs, m^1^A in mRNA accounts for approximately 0.015–0.054% of all adenosines in mammalian cell lines and 0.05–0.16% in mammalian tissues [9,10,21]. m^1^A sites are usually located near the translation start site and the first splice site of mRNA, and they are associated with the translation of coding transcripts [9,10].

In this review, we describe mammalian m^1^A RNA-modifying proteins that specifically install, remove, and bind to the m^1^A base. We also discuss the recent progress in understanding the biological mechanisms involving m^1^A in post-transcriptional gene expression regulation and the biological functions of m^1^A. Further, we review the current approaches for transcriptome-wide and single-base resolution m^1^A detection.

## 2. m^1^A RNA-Modifying Proteins

Reversible m^1^A methylomes in nuclear- and mitochondrial-encoded transcripts are achieved via the dynamic regulation of m^1^A RNA-modifying proteins (m^1^A methyltransferases, m^1^A demethylases and m^1^A-dependent RNA-binding proteins). The characterization of m^1^A-modifying proteins is crucial for understanding the mechanisms underlying m^1^A-mediated gene regulation and the biological roles of m^1^A RNA modification. To date, several m^1^A RNA-modifying proteins responsible for nuclear- and mitochondrial-encoded transcripts have been identified in humans (Figure 1).

### 2.1. m^1^A-Modifying Proteins for Nuclear-Encoded Transcripts

m^1^A is highly abundant in tRNAs, and data on the modifying proteins for m^1^A58 are clearer than those for other m^1^A sites in cytoplasmic tRNAs. The tRNA m^1^A58 methyltransferase complex was first identified in *Saccharomyces cerevisiae,* which contains a non-catalytic subunit and a catalytic subunit encoded by *tRNA (adenine(58)-N(1))-methyltransferase non-catalytic subunit TRM6*
*(Trm6*) and *Trm61*, respectively [22]. Trm61 is critical for AdoMet binding and catalytic function, while Trm6 is responsible for tRNA binding [23]. The conserved homolog of this complex in eukaryotes comprises TRMT61A and TRMT6, which mediate the m^1^A58 modification in cytoplasmic tRNAs [24]. Demethylases for nuclear-encoded tRNAs include α-ketoglutarate-dependent dioxygenase alkB homolog 1 (ALKBH1), ALKBH3 and α-ketoglutarate-dependent dioxygenase FTO (FTO), all belonging to the AlkB family of Fe(II)/α-ketoglutarate-dependent dioxygenases [25,26].

rRNAs act as scaffolds for ribosomal proteins and as ribozymes for peptide bond formation in ribosomes; these functions are partly regulated by post-transcriptional modifications [26]. The m^1^A modification has been founded in the large subunit of nuclear-encoded rRNAs in yeast and humans [27]. Yeast 25S rRNA (adenine(645)-N(1))-methyltransferase (Rrp8) and 25S rRNA (adenine(2142)-N(1))-methyltransferase (Bmt2) were reported to be the methyltransferases responsible for the m^1^A645 and m^1^A2142 modifications, respectively, of 25S rRNA [28,29]. The homolog of yeast Rrp8 in humans is nucleomethylin (NML), a nucleolar factor that catalyzes the m^1^A modification of 28S rRNA in human and mouse cells [30,31]. NML contains a Rossmann-fold methyltransferase-like domain that binds to S-adenosyl-L-methionine, which acts as a methyl donor [30]. YTH domain-containing family protein 3 (YTHDF3) was identified as an m^1^A-dependent RNA-binding protein in HEK293T and RAW264.7 cells, because it showed the highest ability to bind to a human 28S rRNA-derived m^1^A probe. A subsequent RNA immunoprecipitation analysis confirmed that m^1^A-modified RNA was bound by YTHDF3, but not YTHDF1/2, in trophoblast HTR8/SVneo cells [32].

Studies have shown that tRNA m^1^A-modifying proteins are also responsible for the m^1^A modification of mRNAs. For example, the TRMT6/TRMT61A complex mediates the m^1^A addition to mRNA. TRMT6/61A also requires a consensus GUUCRA motif and a tRNA T-loop-like structure for substrate recognition in mRNA, which is similar to tRNA recognition [12]. In contrast, m^1^A in mRNA can be demethylated by ALKBH3 [9,10,33]. In total, 774 m^1^A peaks were reported to be specifically detected in *ALKBH3*-knockout HEK293T cells, and these peaks were enriched in the 5′UTR [9]. Stable isotope labeling by amino acid in cell culture (SILAC)-based quantitative proteomics have identified that YTH domain-containing proteins (YTHDF1-3 and YTHDC1) act as m^1^A-dependent RNA-binding proteins in mRNA; these proteins can directly bind to an m^1^A-modified RNA probe (sequence from the human *transcription factor SOX-18* mRNA) in HEK293T cells. It was also found that Trp432 in YTHDF2 is a key binding residue [34]. m^1^A RNA modification was reported to favor a GA-rich motif, and transcripts containing this motif were increased after ALKBH3 depletion [9,11]. Therefore, the researchers of another study designed a biotin-labeled RNA probe in which m^1^A RNA modification was placed in a GAGGm^1^AG sequence. It could be recognized by YTHDF1 and YTHDF2 in HeLa cells, but YTHDC1 was not detected in this system [35].

### 2.2. m^1^A-Modifying Proteins for Mitochondrial RNAs

tRNA methyltransferase 10 homolog C (TRMT10C) acts as a catalytic core in the TRMT10C/SDR5C1 (also known as HSD17B10, hydroxysteroid 17-β dehydrogenase 10) methylase complex, which is responsible for the m^1^A9 of mt tRNA^Lys^ and m^1^A1374 of mitochondrial *NADH–ubiquinone oxidoreductase chain 5* (*ND5*) mRNA [12,36]. In addition, TRMT10C is also able to catalyze the m^1^G9 modification in mt tRNAs [36]. TRMT61B catalyzes the m^1^A58 of mt tRNA^Leu (UUR)^ and m^1^A947 of mt 16S rRNA [20,37]. Two demethylases, ALKBH1 and ALKBH7, have been shown to remove m^1^A sites from mt RNAs. The level of m^1^A16 of mt-tRNA^Arg^ and m^1^A58 of mt-tRNA^Lys^ was found to be increased upon *ALKBH1* knockout, indicating the role for ALKBH1 in the m^1^A of mt RNAs [38]. The human ALKBH7 has been reported to demethylate m^1^A in Ile and Leu1 pre-tRNA in the mitochondria [39].

## 3. Biological Functions of m^1^A RNA Modification

Since the discovery of m^1^A RNA modification as a chemical modification of RNAs, efforts have been taken to understand the functional characterization of this dynamic methylation in RNA metabolism and gene expression regulation.

### 3.1. m^1^A RNA Modification in RNA Metabolism

m^1^A RNA modification is a pivotal regulator of RNA metabolism, including RNA structure alteration, decay and translation (Figure 2).

The chemical properties of m^1^A RNA modification enable changes in RNA secondary structure. For instance, m^1^A9 and m^1^A58 in tRNAs are required for the conformational shift of mitochondrial tRNA^Lys^ and tRNA^iMet^, respectively, which contribute to the stabilization of alternative native structures [40,41,42,43]. The loss of m^1^A645 has been shown to affect the topological structure of 28S rRNA and alter the RNA interactome [31]. m^1^A was also found to favor the hairpin structure of palindromic RNA sequences, wherein m^1^A can stably localize within apical loops [44]. A recent study revealed that m^1^A RNA modification controlled RNA conformational equilibrium by blocking base-pairing to modulate the RNA duplex [3].

The regulation of m^1^A-modified mRNA decay is mediated by m^1^A-dependent RNA-binding proteins. Limited evidence suggests that the knockdown of YTHDF2 increases the abundance of 7 out of 8 m^1^A-modified transcripts and 2 out of 3 transcripts that bear only the m^1^A but not m^6^A (*N*^6^-methyladenosine) modification [35]. In addition to YTHDF2, YTHDF3 overexpression has been reported to decrease the abundance and decay rate of *insulin like growth factor 1 receptor* (*IGF1R*) mRNA [32].

Translational regulation by m^1^A modification varies among different RNA types. The m^1^A demethylases ALKBH1 and FTO have been reported to control specific tRNA m^1^A demethylation and decrease translation initiation [45,46]. Eukaryotic elongation factor 1-α (eEF1α) immunoprecipitation was used to reveal that m^1^A-methylated tRNAs are enriched in polysomes, indicating the role of m^1^A RNA modification in translation activation [45]. During retroviral reverse transcription in early human immunodeficiency virus 1 (HIV-1) replication, TRMT6-mediated m^1^A58 of tRNA_3_^Lys^ acted as a stop site that contributed to genome integration [47]. Further, mRNAs carrying m^1^A undergo translation repression because of interfered Watson–Crick base pairing [8,12,48].

### 3.2. m^1^A RNA Modification in Biological Processes

Post-transcriptional modifications are involved in various biological processes, and recent evidence showed the importance of m^1^A RNA modification in this field. In a high-temperature-sensitive *Thermococcus kodakarensis* strain, decreased m^1^A58 and melting temperature of tRNA were observed, suggesting the relevance of m^1^A58 and the growth ability of this strain at high temperatures [49]. m^1^A RNA modification was found to exhibit its protective ability of RNAs under stress conditions. During heat shock, m^1^A-harbouring transcripts were found to preferentially accumulate in stress granules, subsequently resulting in a shorter time to restore the translation state during recovery [50]. Alkylating agents induced m^1^A modification in RNAs and orchestrated translational suppression by recruiting the ASCC damage repair complex (activating signal cointegrator 1 complex) [51]. The tRNA modification profiles of the *Aplysia* central nervous system showed increased m^1^A RNA modification levels in animals after behavioral training [52]; this was the first study to characterize the variable pattern of m^1^A RNA modification during defensive reflex-associated behavioral sensitization. *Petunia* TRMT61A catalyzed m^1^A RNA modification in mRNAs, and the knockdown of TRMT61A decreased the chlorophyll content and changed chlorotic and wrinkled leaf phenotype [53]. A recent study showed that the m^1^A demethylase ALKBH3 functioned as a negative regulator of ciliogenesis by removing the m^1^A sites on *Aurora A* mRNA (a key regulator of cilia disassembly) in mammalian cells, which was further involved in cilia-associated developmental processes in zebrafish [54].

## 4. m^1^A RNA Modification in Diseases

The limited exploration of m^1^A RNA modification as a pathological feature has mainly focused on tumor progression (Table 1). It was reported that the knockdown of m^1^A demethylase ALKBH3 increased the abundance of m^1^A RNA modification in small RNAs (< 200 nucleotides) along with suppressed nascent protein in pancreatic cancer cells [55]. The ALKBH3-dependent m^1^A demethylation of macrophage *colony-stimulating factor 1* (*CSF1*) mRNA enhanced its mRNA stability and thus promoted the invasion of breast and ovarian cancer cells [56]. In addition, ALKBH3 removed the m^1^A RNA modification of tRNA^GlyGCC^ to promote tRNA cleavage by angiogenin. The generation of excessive tRNA-derived small RNAs may affect ribosome assembly and apoptosis in HeLa cells [57]. Furthermore, ALKBH3 promoter CpG island hypermethylation and transcriptional silencing were found in Hodgkin lymphoma cells, which were identified as a potential prognostic biomarker associated with poor clinical outcomes in patients with Hodgkin lymphoma [58]. A recent study found that levels of tRNA m^1^A modification were upregulated in hepatocellular carcinoma (HCC) tissues. The TRMT6/TRMT61A complex mediated increased m^1^A58 levels in tRNA, which then triggered *peroxisome proliferator-activated receptor delta* (*PPARδ*) mRNA translation in HCC stem cells. PPARδ promoted cholesterol biogenesis to activate the Hedgehog pathway, thereby initiating the self-renewal of HCC stem cells [59].

## 5. Approaches for m^1^A RNA-Modification Detection

Several methods have been developed for the detection of m^1^A modifications in RNAs. m^1^A can be detected by bi-dimensional thin-layer chromatography (2D-TLC) or high-pressure liquid chromatography (HPLC), both of which utilize the differential retention properties for the separation of various nucleosides [4]. Furthermore, the coupling of liquid chromatography to mass spectrometry (LC–MS/MS) can provide a more sensitive approach for quantifying the m^1^A level in RNAs [4]. The drawback of these methods is that the location information of m^1^A is generally lost.

Recent advances in high-throughput technologies have revealed the global m^1^A distribution map and dynamics in the transcriptome, and developed tools provide easy ways for m^1^A site-specific detection. Here, we summarize the high-throughput sequencing approaches for m^1^A RNA-modification detection (Table 2).

### 5.1. m^1^A Antibody-Independent Detection Methods

Several enzyme-based strategies have been developed for detecting m^1^A RNA modification. *Escherichia coli* α-ketoglutarate-dependent dioxygenase, AlkB, can demethylase several post-translational modifications to avoid unexpected termination during reverse transcription. In ARM-seq (AlkB-facilitated RNA methylation sequencing), the removal of m^1^A, m^3^C (*N*^3^-methylcytidine), or m^1^G (*N*^1^-methylguanosine) modifications by AlkB treatment facilitates the production of full-length cDNAs from previously modified templates, producing a ratio of reads in treated versus untreated samples that can be used to identify methylated RNAs [7]. RT-1306 is an evolved HIV-1 reverse transcriptase that can induce m^1^A-specific base mutations, thus providing a direct and rapid method for mapping m^1^A modification [11]. One possible drawback is that the accuracy is dependent on the fidelity and specificity of the enzymes used in these approaches.

### 5.2. m^1^A Antibody-Dependent Detection Methods

m^1^A antibody-dependent RNA immunoprecipitation has several advantages in m^1^A-modification detection. m^1^A antibody-dependent methylated RNA immunoprecipitation sequencing (m^1^A-MeRIP-seq) combined with AlkB demethylation (m^1^A-ID-seq) has been used to describe m^1^A dynamics in the human transcriptome [9]. Dominissini and colleagues also developed a method termed ‘m^1^A-seq’ that combined MeRIP-seq and Dimroth rearrangement to obtain the mRNA m^1^A methylome in human cells [10]. Moreover, combining advanced reverse transcriptase TGIRT (thermostable group II intron reverse transcriptase) and SuperScript III—which enable efficient RT misincorporation and truncation at m^1^A sites—with m^1^A antibody was shown to provide a feasible way to map m^1^A at a single-base resolution (m^1^A-seq-TGIRT and m^1^A-seq-SS) [12]. The misincorporation-assisted profiling of m^1^A (m^1^A-MAP) has improved the previous m^1^A-ID-seq by utilizing TGIRT and AlkB treatment. m^1^A-MAP has been used to identify a total of 740 m^1^A sites in the transcriptome and showed a GUUCRA consensus motif located in a subset of tRNA substrates of the TRMT6/61A complex [8]. A major drawback of these approaches is the non-specific binding of m^1^A antibodies, which may cause false positives.

### 5.3. Site-Specific m^1^A Detection

Based on m^1^A-induced Watson–Crick base-pairing disruption, some approaches have been developed for site-specific m^1^A evaluation. m^1^A induces the truncation or mutation of cDNA during reverse transcription, so primer extension can be applied to compare m^1^A modification levels based on the intensity of the truncated and full-length cDNA bands [60]. The clustered regularly interspaced short palindromic repeat (CRISPR)-associated endoribonuclease Cas13 can mediate the cleavage of both target and collateral RNAs after the formation of the Cas13/crRNA/target RNA complex. Based on this phenomenon, a reporter system with CRISPR/Cas13a can produce a high-fluorescence signal by cleaving quenched fluorescent RNAs with the correct Watson–Crick base-pairing of crRNA or target RNA but not m^1^A-modified RNAs [61]. Another method takes advantage of the different nick ligation efficiencies by T3 DNA ligases at the adenosine and m^1^A base. It can easily detect the m^1^A state at a specific site by analyzing the PCR amplification products from nick ligation [62]. These methods are suitable for RNA with high abundance and known sequence of m^1^A sites rather than the identification of novel modification sites.

## 6. Future Perspectives

Although findings related to the biological functions of the m^1^A modification have been reported, some key questions remain to be addressed. For instance, studies on m^1^A-dependent RNA-binding proteins are limited. With the advances in the sensitivity and throughput of proteomics techniques, it is critical to further characterize m^1^A-dependent RNA-binding proteins involved in m^1^A recognition and RNA metabolism, as well as to determine how they recognize the m^1^A sites in their target transcripts. In addition, the reported methyltransferases and demethylases may target different transcripts in different cell types or biological processes; therefore, the context-dependent roles and regulatory mechanisms of m^1^A-modifying proteins are also interesting and important to explore. Further, optimized technologies are still needed for transcriptome-wide and single-base resolution m^1^A detection. These will greatly improve our understanding of the biological roles of m^1^A in RNA. Finally, more work is needed to understand the roles of the m^1^A RNA modification in human diseases.

## Figures and Tables

**Figure 1 genes-13-00910-f001:**
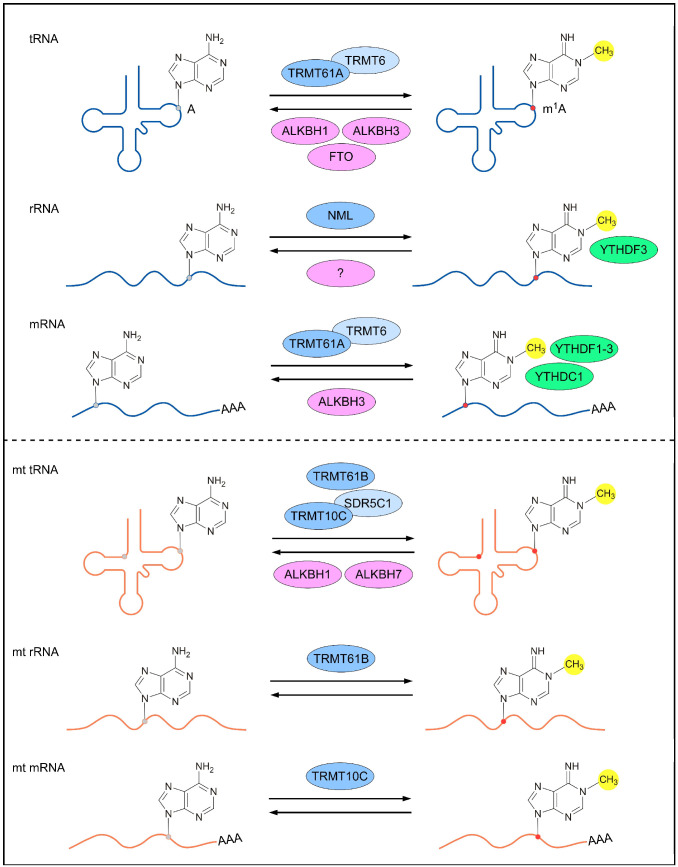
m^1^A-modifying proteins for different types of RNAs. The nuclear-encoded (top panel) and mitochondrial (bottom panel) RNAs are reversibly methylated by m^1^A methyltransferases (blue; dark blue represents catalytic core of the methylase complex), demethylased by m^1^A demethylases (pink), and bound by m^1^A-dependent RNA-binding proteins (green). A, adenosine; m^1^A, *N*^1^-methyladenosine; TRMT, tRNA (adenine (58)-N (1))-methyltransferase subunit; ALKBH, α-ketoglutarate-dependent dioxygenase alkB homolog; FTO, α-ketoglutarate-dependent dioxygenase alkB homolog FTO; NML, nucleomethylin; YTHDF, YTH domain-containing family protein; YTHDC1, YTH domain-containing protein 1; SDR5C1, 3-hydroxyacyl-CoA dehydrogenase type-2.

**Figure 2 genes-13-00910-f002:**
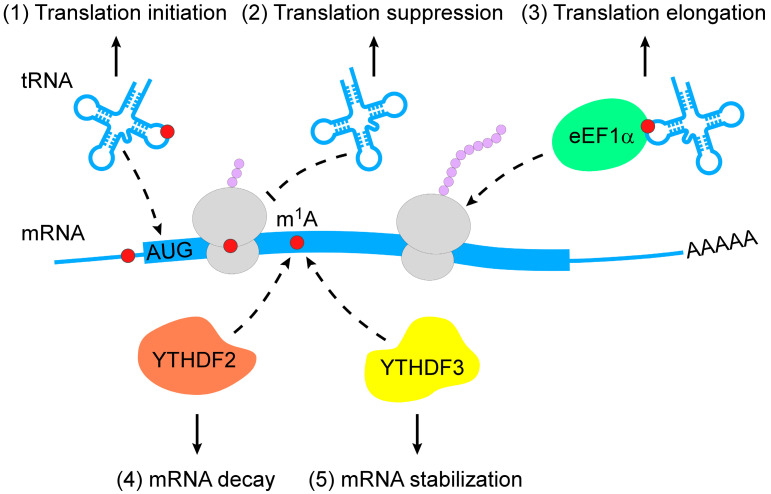
Action mechanisms of m^1^A in RNA metabolism. m^1^A RNA modification regulates RNA metabolism in multiple layers (from top to bottom: (1) m^1^A RNA modification stabilizes tRNAs to promote translation initiation; (2) m^1^A-modified mRNAs interfere with Watson–Crick base-pairing with tRNA to suppress translation; (3) m^1^A-modified tRNAs are coupled with eEF1α to polysomes to promote translation elongation; (4) m^1^A-modified mRNAs are subjected to degradation by interacting with YTHDF2; (5) m^1^A-modified mRNAs become stable when they bind to YTHDF3). m^1^A, *N*^1^-methyladenosine; eEF1α, eukaryotic elongation factor 1-α; YTHDF, YTH domain-containing family protein.

**Table 1 genes-13-00910-t001:** Dysregulation of m^1^A RNA modification in human cancers.

Cancers	m^1^A-Modifying Proteins	Roles	Targets	Mechanisms	Refs
Pancreatic cancer	ALKBH3	Oncogene	small RNAs	Unknown	[55]
Breast and ovarian cancer	ALKBH3	Oncogene	CSF1	mRNA decay	[56]
Cervical cancer	ALKBH3	Oncogene	tRNAs	tRNA cleavage	[57]
Hodgkin lymphoma	ALKBH3	Tumor suppressor	COL1A1, COL1A2	Unknown	[58]
Hepatocellular carcinoma	TRMT6/TRMT61A	Oncogene	tRNAs	Unknown	[59]

ALKBH, α-ketoglutarate-dependent dioxygenase alkB homolog; TRMT, tRNA (adenine(58)-N(1))-methyltransferase subunit; CSF-1, macrophage colony-stimulating factor 1; COL1A1, collagen α-1(I) chain; COL1A2, collagen α-2(I) chain.

**Table 2 genes-13-00910-t002:** Detection methods for mapping m^1^A RNA modification.

Methods	Features	RNA Substrate	Antibody Dependent	Resolution	Cell Line
ARM-seq [7]	AlkB treatment	tRNA	No	Fragment	BY4741, GM05372, GM12878
m^1^A-quant-seq [11]	Spike-in m^1^A RNA; AlkB treatment; RT-1306-mediated RT mutation	tRNA, rRNA, mRNA, lncRNA	No	Single base	HEK293T
m^1^A-seq [10]	m^1^A-RNA immunoprecipitation; Dimroth rearrangement	rRNA, mRNA	Yes	Fragment	BY4741, Sp1, MEF, mESC, HeLa
m^1^A-ID-seq [9]	m^1^A-RNA immunoprecipitation; AlkB treatment; AMV-mediated RT truncation	rRNA, mRNA, lncRNA	Yes	Fragment	HEK293T
m^1^A-seq-SS [12]	m^1^A-RNA immunoprecipitation; Dimroth rearrangement; SS-mediated RT truncation	tRNA, rRNA, mRNA, lncRNA	Yes	Single base	HEK293T
m^1^A-seq-TGIRT [12]	m^1^A-RNA immunoprecipitation; Dimroth rearrangement; TGIRT-mediated RT readthrough	tRNA, rRNA, mRNA, lncRNA	Yes	Single base	HEK293T
m^1^A-MAP [8]	m^1^A-RNA immunoprecipitation; AlkB treatment; TGIRT-mediated RT read-through	tRNA, rRNA, mRNA, lncRNA	Yes	Single base	HEK293T
m^1^A-IP-seq [11]	m^1^A-RNA immunoprecipitation; AlkB treatment; RT-1306-mediated RT mutation	tRNA, rRNA, mRNA, lncRNA	Yes	Single base	HEK293T

AlkB, α-ketoglutarate-dependent dioxygenase; RT, reverse transcription; AMV, avian myeloblastosis virus; SS, SuperScript III; TGIRT, thermostable group II intron reverse transcriptase; BY4741, *saccharomyces cerevisiae* BY4741 strains; GM05372, human B lymphocyte-derived GM05372 cell; GM12878, human B lymphocyte-derived GM12878 cell; HEK293T, human embryonic kidney 293T cell; Sp1, *schizosaccharomyces pombe* Sp1 cells; MEF, mouse embryonic fibroblasts; mESC, mouse embryonic stem cell; HeLa, human HeLa cell.

## Data Availability

Not applicable.

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
