# Peer review of "m1A RNA Modification in Gene Expression Regulation"

_genes, 2022, doi:10.3390/genes13050910_

Round 1
Reviewer 1 Report
The review of Hao Jin et al. is focussed on m1A RNA modification and its role in gene expression. Some references are missing and Figure 1 should be modified.
Figure 1 is misleading for TRMT10C: it gives the feeling that TRMT10C is responsible for the m1A58 and not M1A9. It should be added that TRMT10C is also able to catalyze the m1G9 modification in tRNAs.
About the m1A modification of mt tRNAs, the recent paper of Suzuki et al. Nat Comm 2020 should be cited and the number of m1A modifications and the concerned mt tRNA must be correct accordingly this paper.
l28: m1A modification in tRNA are also found for nucleotide A16 in mt tRNA, for A22 in bacteria and for A57 as a intermediate of I57. This sentence must be corrected and references added. This is reviewed in Oerum et al. biomolecules 2017 for instance.
l150: Post-translational is an error, post-transcriptional is the right word.
Author Response
Author's Reply to the Review Report (Reviewer 1)
The review of Hao Jin et al. is focused on m1A RNA modification and its role in gene expression. Some references are missing and Figure 1 should be modified.
1. Figure 1 is misleading for TRMT10C: it gives the feeling that TRMT10C is responsible for the m1A58 and not M1A9. It should be added that TRMT10C is also able to catalyze the m1G9 modification in tRNAs.
Response: We appreciate the reviewer's instructive suggestions. According to the reviewer's valuable suggestion, we have added m1A9 in Figure 1. Since Figure 1 is focused on the modifying proteins for m1A, we did not include m1G9 in the figure; however, we have added the description “TRMT10C is also able to catalyze the m1G9 modification in mt-tRNAs” in the revised manuscript (lines 118).
2. About the m1A modification of mt tRNAs, the recent paper of Suzuki et al. Nat Comm 2020 should be cited and the number of m1A modifications and the concerned mt tRNA must be correct accordingly this paper.
Response: We thank you for the helpful suggestion. We have cited the paper (Suzuki et al. Nat Comm 2020) and updated the descriptions of the number of m1A modifications and the concerned mt-tRNA in our revised manuscript (lines 33-42).
3. Line 28: m1A modification in tRNA are also found for nucleotide A16 in mt tRNA, for A22 in bacteria and for A57 as a intermediate of I57. This sentence must be corrected and references added. This is reviewed in Oerum et al. biomolecules 2017 for instance.
Response: According to your suggestion, we have corrected this sentence in our revised manuscript (lines 33-42) and added the references (Oerum et al. biomolecules 2017; Suzuki et al. Nat Comm 2020).
4. Line 150: Post-translational is an error, post-transcriptional is the right word.
Response: We apologize for this oversight. We have corrected “post-translational” to “post-transcriptional” in our revised manuscript (line 163).
Reviewer 2 Report
Overall the review was well written, easy to follow and was very inclusive. I like the attempt of the authors to include the role of m1A and associated enzymes/proteins in tRNA, rRNA, and mRNA. The RNA modification field has been growing as interest in mRNA modifications grows, and this is a simple review that will aid any researcher or student interested in the field.
With that there are a few issues that need to be corrected:
The first error that I would like to point out is a common misconception to researchers new to the RNA modification field. m1A and m3C do not have a positive charge in a neutral pH environment. The correct structures can be easily looked up in the modomics database. Please do not perpetuate this misconception.
Throughout the paper the authors confuse the reader by using the terms "writers", "erasers" and "readers". These are non-sense terms that do not have any real meaning. The enzymes/proteins in question are N1-adenosine methyltransferases (or simply methyltransferase), demethylases, and m1A dependent RNA binding proteins. Do not confuse the reader with different terms that are not precise and have very little meaning.
I think it would be beneficial to discuss the drawbacks to the different detection methods used to detect m1A in RNA. Also the review completely ignores the historical methods such as TLC, and especially the gold standard, mass spectrometry. While RNA mass spectrometry cannot be performed with the throughput of next-gen sequencing, the technique certainly warrants mention.
Author Response
Author's Reply to the Review Report (Reviewer 2)
Overall the review was well written, easy to follow and was very inclusive. I like the attempt of the authors to include the role of m1A and associated enzymes/proteins in tRNA, rRNA, and mRNA. The RNA modification field has been growing as interest in mRNA modifications grows, and this is a simple review that will aid any researcher or student interested in the field. With that there are a few issues that need to be corrected:
1. The first error that I would like to point out is a common misconception to researchers new to the RNA modification field. m1A and m3C do not have a positive charge in a neutral pH environment. The correct structures can be easily looked up in the modomics database. Please do not perpetuate this misconception.
Response: We thank the reviewer for raising this important concern. According to your suggestion, we have revised the description in our revised manuscript (lines 21-22) as “ N1-methyladenosine (m1A) is a reversible methylation involving the addition of a methyl group at the N1 position of adenosine in cellular RNA transcripts” and “The secondary structure and RNA-protein interaction of m1A-modified RNAs are also altered under physiological conditions” (lines 24-26).
2. Throughout the paper the authors confuse the reader by using the terms "writers", "erasers" and "readers". These are non-sense terms that do not have any real meaning. The enzymes/proteins in question are N1-adenosine methyltransferases (or simply methyltransferase), demethylases, and m1A dependent RNA binding proteins. Do not confuse the reader with different terms that are not precise and have very little meaning.
Response: We thank you for the kind suggestion. We have changed the terms "writers", "erasers", and "readers" to “m1A methyltransferases”, “m1A demethylases”, and “m1A-dependent RNA binding proteins, respectively, in our revised manuscript.
3. I think it would be beneficial to discuss the drawbacks to the different detection methods used to detect m1A in RNA. Also the review completely ignores the historical methods such as TLC, and especially the gold standard, mass spectrometry. While RNA mass spectrometry cannot be performed with the throughput of next-gen sequencing, the technique certainly warrants mention.
Response: We thank you for the helpful suggestion. We have added some information on the historical methods, such as TLC and mass spectrometry, in the revised manuscript (lines 206-211). In addition, we have discussed the drawbacks of the different detection methods used to detect m1A in RNA in Section 5 “Approaches for m1A RNA modification detection”.
Reviewer 3 Report
Dear Author,
- The introduction part is insufficient. It needs to be elaborated.
- Ommit old references.
- The manuscript needs to be edited by native English speaker for English Language and Grammar.
Author Response
Author's Reply to the Review Report (Reviewer 3)
1. The introduction part is insufficient. It needs to be elaborated.
Response: We thank you for the helpful suggestion. We have added more descriptions of m1A in the introduction section of our revised manuscript.
2. Ommit old references.
Response: According to your suggestion, we have replaced old references with new references in our revised manuscript (Reference section).
3. The manuscript needs to be edited by native English speaker for English Language and Grammar.
Response: According to the reviewer's kind suggestion, our revised manuscript has been carefully proof-read by a native English speaker from a professional manuscript editing service.
Round 2
Reviewer 3 Report
Dear Author,
The rebuttal against the queries are satisfactory.